# Irrigation suitability, health risk assessment and source apportionment of heavy metals in surface water used for irrigation near marble industry in Malakand, Pakistan

**Asghar Khan** [1] *, **Muhammad Saleem Khan**[1], **Juan José Egozcue**[2], **Munib Ahmed Shafique** [3], **Sidra Nadeem**[3], **Ghulam Saddiq**[4]

1 Department of Botany, Islamia College Peshawar, Peshawar, Khyber Pakhtunkhwa, Pakistan, 2 Department of Civil and Environmental Engineering, Technical University of Catalonia, Barcelona, Spain, 3 Pakistan Institute of Nuclear Science and Technology Islamabad, Islamabad, Pakistan, 4 Department of Physics, Islamia College Peshawar, Peshawar, Khyber Pakhtunkhwa, Pakistan

* malakand35@gmail.com

**Data Availability Statement:** All relevant data are within the paper and its Supporting Information files.

## Abstract

Water is a vital, finite resource whose quantity and quality are deteriorating as the world population increases. The current study aims to investigate the concentration of heavy metals (HM) in surface water for irrigation purposes with associated human health risks and pollution sources near the marble industry in Malakand, Pakistan. Twenty-seven water samples were randomly collected and analyzed for HM concentration by inductively coupled plasma–optical emission spectrometry (ICP–OES). pH, electrical conductivity (EC), total dissolved solids (TDS), biological oxygen demand (BOD), and chemical oxygen demand (COD) were measured using standard methods of American Public Health Association (APHA). Irrigation suitability was assessed using specific water quality parameters. The associated health risks from ingestion and dermal exposure to heavy metals were calculated by USEPA health risk indices. Pollution sources and spatial distribution mapping were studied using compositional data analysis (CoDa) and the application of a geographic information system (GIS) to understand the changing behavior of heavy metals in surface waters. The concentrations of BOD (89%), COD (89%), Al (89%), Ca (89%), Cr (56%), Cu (78%), Fe (56%), K (34%) Mg (23%), Mn (56%), Na (89%), Ni (56%), P (89%), and Zn (11%) exceeded the safety limits of National Environmental Quality standards (NEQs) of Pakistan. The results of Kelly's ratio (KR) classified surface water as unsuitable for irrigation. The average daily doses (ADD, mg/kg/day) for Al, Cu, Cr, Fe, Mn, Ni, and Zn were higher in children than in adults. The hazard index (HI) for children and adults was above the threshold (HI > 1), indicating a significant risk of non-carcinogenic toxicity. The carcinogenic risk values for Cr and Ni were above the USEPA limit ($1 \times 10^{-6}$ to $1 \times 10^{-4}$), suggesting a potential carcinogenic risk for the target population. Principal component analysis (PCA), biplot (CLR), and the CoDa-dendrogram allowed for the identification of elemental associations, and their potential source was anthropogenic rather than natural in origin. Regular monitoring and phytoremediation strategies are proposed to safeguard crops and human health.

**Funding:** The author(s) received no specific funding for this work.

**Competing interests:** The authors have declared that no competing interests exist.

## Introduction

Water is a vital, finite resource whose quantity and quality are deteriorating with an increase in world population [1]. The process of industrialization and urbanization is beneficial, especially in terms of shaping demographic characteristics. However, rapid, unplanned, and uncontrolled changes in the physical landscape can lead to degradation and scarcity of water resources [2, 3]. The agricultural sector consumes approximately 70% of all water resources, and by 2025, almost 60% of the world population could face physical water stress from over-consumption [4]. The increasing demand for water to meet the food needs of a growing population is one of the main reasons for reusing wastewater for urban and peri-urban irrigation [5]. Reusing wastewater for irrigation can alleviate the intense pressure on freshwater abstraction and provide nutrients to plants and crops [6]. Without proper management, a significant amount of wastewater can affect water availability, general economic conditions, quality of life, and even the cultural and religious values of human society [7].

Heavy metal (HM) pollution and the associated safety of freshwater reservoirs and human health are major global concerns today [8, 9]. HM such as arsenic (As), chromium (Cr), copper (Cu), cadmium (Cd), mercury (Hg), nickel (Ni), lead (Pb), and zinc (Zn) are major pollutants in freshwater reservoirs. Heavy metals are persistent in the environment and nonbiodegradable [10]. Heavy metals can enter and bioaccumulate the human body through a series of food chains, eventually leading to kidney disease, hormonal imbalances, hair loss, cardiovascular failure, neurological and endocrine disorders, cancer, and respiratory and digestive problems [11]. In addition, the distribution of heavy metals in aquatic environments affects the biotic community through biogeochemical interactions and the formation of pollutant complexes from organic matter [12]. A long-term follow-up study showed that using contaminated water to irrigate agricultural land increases the relative risk, leading to the accumulation of heavy metals in the soil and tissues of food crops that support the living system [13]. Monitoring and assessment of heavy metals in water, soil, and food crops are important aspects considered to protect public health [14].

Pakistan is the sixth largest nation in the world, with a population of approximately 208 million, and is expected to surpass the 240 million mark by 2030 [15]. This increase in population can have several drastic impacts on freshwater availability, and water demand is projected to far exceed supply [16]. Pakistan, formerly known as a water surplus country, is currently suffering from water scarcity as the country is consuming its available freshwater resources at a high rate. In Pakistan, water availability has dropped from 1299 m$^3$ per capita to 1100 m$^3$ in 1996–2006 and is projected to be less than 700 m$^3$ per capita by 2025 compared to the international standard of 1500 m$^3$ [17]. In Pakistan, industries such as textile, steel, sugar, food processing, tanneries, chemicals, refineries, and petrochemicals are the main contributors to water pollution. These industries produce several thousand gallons of wastewater containing vast amounts of potentially toxic metals such as Hg, Pb, Cd, Cr, Ni, Fe, Cu, and Co [18]. According to one estimate, more than 7.5708 $\times 10^9$ liters of wastewater are discharged into water drains daily without prior treatment [15].

Almost 70% of marble's valuable mineral resources are lost during the quarrying, processing, and refining process, and almost 40% of marble waste is released into the surrounding atmosphere, leading to widespread environmental pollution [19, 20]. Processing a ton of marble releases almost a ton of slurry, which is 70% water and 30% powder [21]. After some retention in a sedimentation tank, water in the slurry is reused. However, the waste marble powder in the slurry remains a source of contamination for water, soil, and air [22] The direct discharge of the untreated marble effluents into nearby streams and rivers alters its physicochemical properties. Overall, it causes problems for farmers, fish farmers, pastoralists and the general public, as water is unsuitable for irrigation and drinking purposes [23, 24].

Wastewater treatment can improve the quality of industrial effluents, but financial constraints and the implementation of effective waste management policies are major challenges for proper quality control in developing countries [25, 26]. In Pakistan, wastewater collection and treatment schemes at the secondary or tertiary level are lacking, and only a fraction ($< 8\%$) of industrial wastewater is properly treated [27]. The rest is discharged directly into rivers, streams, irrigation canals, and other bodies of water without mandatory treatment per environmental protection agency requirements [3]. Protecting the availability of sufficient amounts of high-quality water is a major technological and societal challenge that needs to be addressed [28, 29]. In Pakistan, the concept of using constructed wetlands to remove toxic metals from wastewater was recently introduced [30]. The use of biochar is also one of the most advanced techniques for removing toxic compounds from water and soil [31]. In addition, biological and chemical remediation technologies such as plant growth-promoting rhizobacteria (PGPR) and the use of diatomite are effective in heavy metal remediation [32, 33].

In developing countries, including Pakistan, poor farmers often rely on the use of wastewater to grow various crops for short-term economic benefits, ignoring the environmental and related health risks due to a lack of basic knowledge [3]. Wastewater contains pathogens (bacteria, viruses, protozoa, and molds) and potentially toxic elements in excessive concentrations that can cause significant health problems [34, 35]. Exposure to wastewater can occur through ingestion, dermal, and inhalation routes while working in wastewater-generating industries, using wastewater for irrigation and swimming [36, 37]. Therefore, the use of wastewater for irrigation and household purposes can pose short- and long-term risks [34]. In the Malakand Division, the marble industry generates waste in a variety of forms that are discharged directly into rivers and irrigation canals, rendering the surface water unsuitable for irrigation and household purposes [23]. This increasing trend of surface water pollution has become a serious problem [38]. Health risk assessments such as hazard index and cancer risk could be used to determine noncarcinogenic and carcinogenic health risks from exposure to potentially toxic elements through ingestion, inhalation, and dermal contact [39].

Similarly, in large industrial areas where point and diffuse sources of pollution are mixed, the sources are difficult to distinguish. Therefore, different approaches to defining geochemical baselines are required [40]. The geochemical factors associated with water, soil, and sediments should be considered compositional data, as they contain valuable information based on elemental ratios [41]. In the case of sole interest in the relative structure of geochemical information, the application of classical statistical tools to the input concentrations can lead to spurious results, since compositional data follow different geometric rules [42]. It is well known that traditional statistical methods based on Euclidean distance and covariance structure are not relevant for geochemical concentrations, leading to a spurious correlation of element associations [43, 44]. In brief, geochemical data are positive vectors that are usually represented as summing to a constant value [43] and are different from those in real Euclidean space [45]. A family of log-ratio transformations, such as additive log-ratio (ALR), centered log-ratio (CLR), and isometric log-ratio (ILR), can handle element concentration data that are inherently compositional [46–48]. The centered log-ratio (CLR) transformation provides a compositional alternative to raw data that uses the average of all variables but results in collinearity, while the isometric log-ratio (ILR) transformation avoids collinearity [48, 49]. The centered log-ratio (CLR) transformation developed by [44] was applied to the data set in combination with compositional data analysis (CoDa) principles such as the variation matrix, CLR biplot, and CoDa dendrogram to establish a linear relationship between the CLR variables and differentiate the source of geochemical compounds [50–52]. In addition, it is important to identify the unusual patterns that might be related to heavy metal pollution in an area of interest. Traditional geochemical modelling methods mainly focus on the statistical aspects of

geochemical data and neglect the spatial distribution of geochemical samples [53]. Therefore, both spatial and statistical aspects of geochemical anomalies are considered. Likewise, exposure to chemical elements through ingestion and dermal route, assessment, and monitoring of water quality is critical to understanding, protecting, and managing environmental resources and human health and well-being [54]. In conjunction with the above context, the current study aimed (i) to investigate heavy metal pollution in surface water near marble processing plants. (ii) To evaluate the associated health risks from exposure to surface water through ingestion and dermal contact and (iii) to identify potential sources of heavy metal contamination in surface water by compositional data analysis (CoDa) and spatial distribution mapping.

## Material and methods

### Description of the study area

The study area lies within geographic coordinates 35N and 72E (Fig 1), which represent a gateway to the districts of Swat, Dir, and Chitral and the merged tribal districts of Bajaur and Mohmand. It is bounded to the north by the Swat foothills, to the south by the Mohmand Melange complex and Charsadda District, and to the west by the Kot Melange complex [55]. Malakand District covers a total area of 952 km$^2$ with a population density of 475 people per km$^2$ [56]. Climatically, the area falls into the subtropical and humid temperate zone, which is dominated by chir pine, olive, oak, and acacia species [57]. The study area receives an average annual

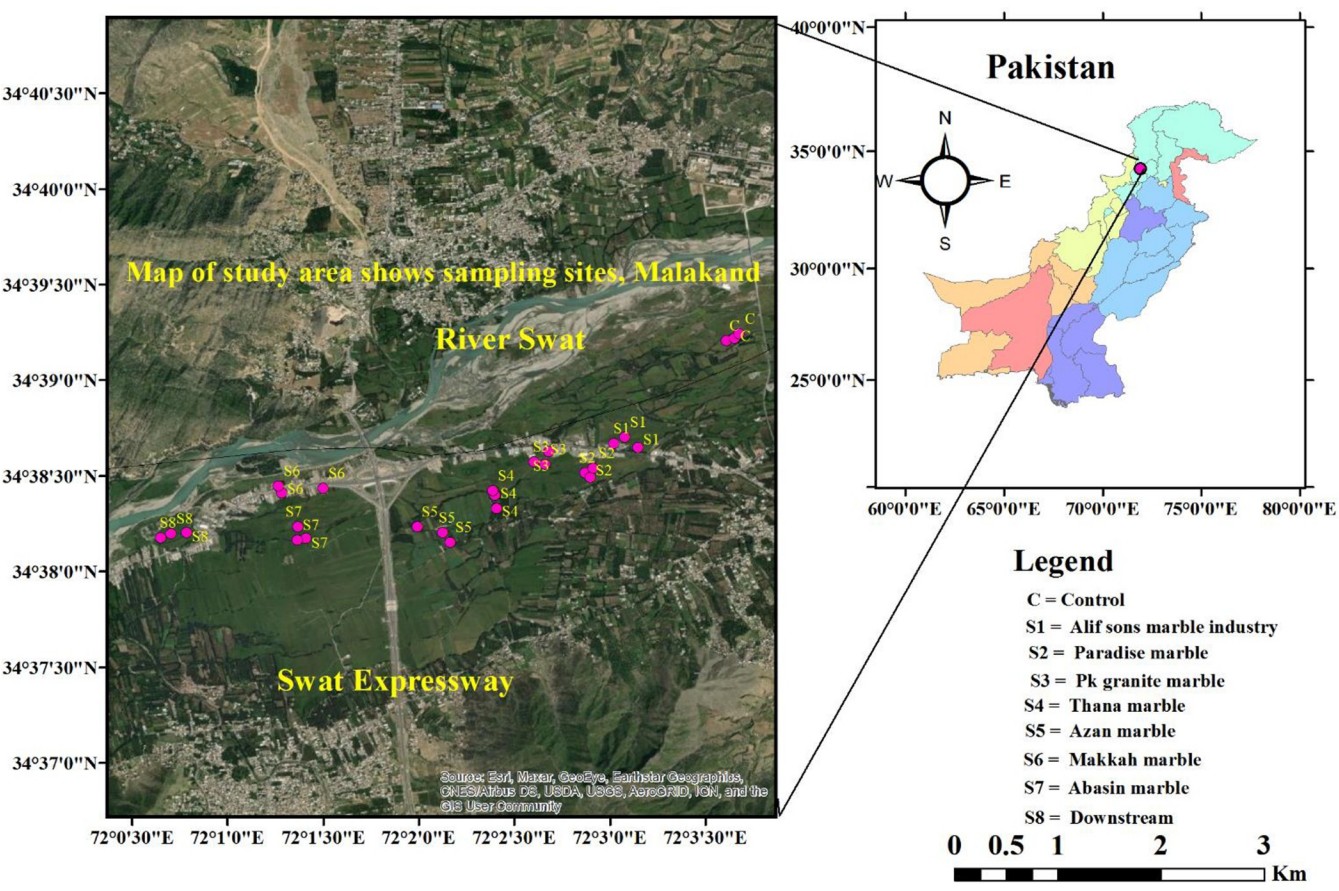

**Fig 1. The map of the study area shows the sampling sites.**

rainfall of 600 to 650 mm. The maximum summer temperature reaches 41.9˚C, while the minimum winter temperature can drop to –2˚C. The hottest months of the year are June and July, with average temperatures of 40˚C and 38˚C, respectively [58].

Geologically, the area studied consisted of alluvial deposits, Malakand granite, Chakdara granite, Dir metasediments, Stream deposits, Kashala formation, Peshmal schist, Indus suture melange, and Kohistan batholith and granite formation. These deposits are enriched with valuable minerals, and rocks contain a combination of graphitic phyllite with garnet grains finely overlaid with marble beds [38]. Malakand granite, Dargai chromite, marble, mica, quartz, and other valuable mineral resources have been reported at various geographic points in the study area [59].

## Experimental site selection

The Marble Processing Plants (MPP) in the study area discharge their effluents directly into the nearby irrigation canals originating from the Swat River (Fig 1). Surface runoff, municipal wastewater, and atmospheric deposition of heavy metals further contribute to irrigation canal pollution in the study area [60]. The people of the study area are primarily associated with the agricultural sector and grow various crops and vegetables on the river water [61]. Therefore, the quality of the irrigation water and the associated health risks are extremely important due to the daily exposure to contaminated surface water. Considering this, a preliminary field survey of the irrigation canals near the Marble Processing Plants in Malakand, District of Pakistan, was conducted (Fig 2A and 2B). Irrigation channels derived from the river were observed for at least six months from April 2019 to August 2019. The study area was divided into three main sites: (i) upstream control (C), which is lacking in the MPP; (ii) industrial sites (S1-S7), where the MPP is operational; and (iii) downstream (S8), which collects most of the marble wastewater and is distant from the MPP (Fig 1). The coordinates of the sampling points were recorded by GPS (GERMIN ETREX10, China). A map of the study area (Fig 1) was generated using ArcGIS version 10.2.2.

## Collection of samples of surface water

From September to December 2019, twenty-seven samples of surface water used for irrigation and household purposes were collected in clean 500 mL polyethylene bottles at the selected

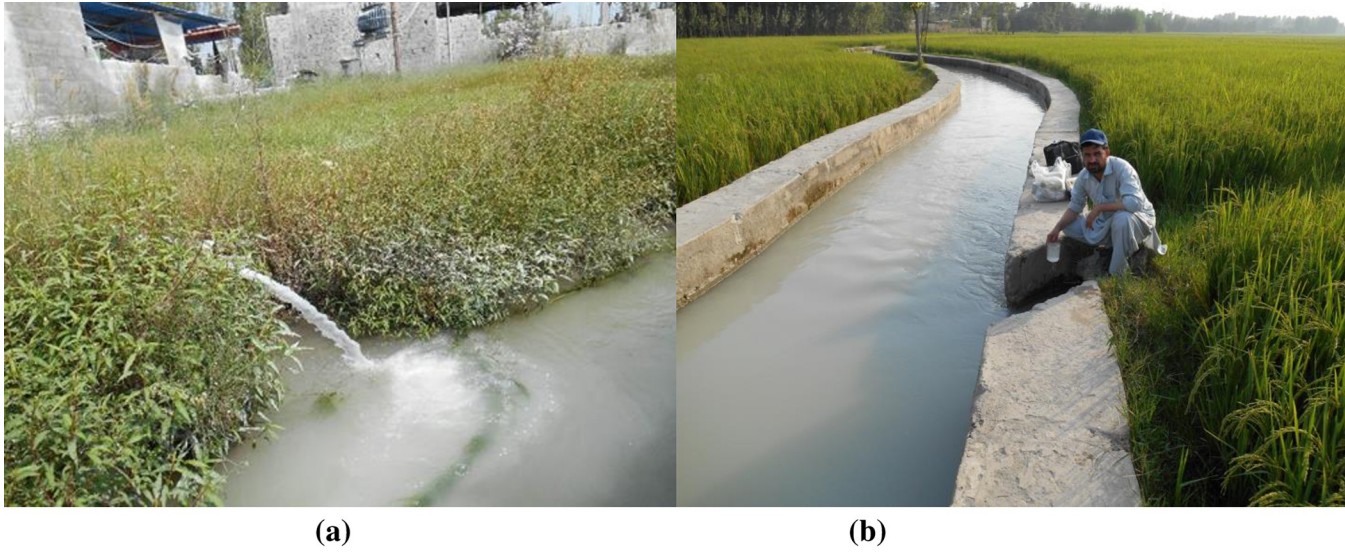

**(a)**                    **(b)**

**Fig 2.** (a) Effluent discharge from the marble industry into the irrigation canal (b) Water sample collection by the author (Asghar Khan) from the study area.

sites (Fig 1). Before sampling, the bottles were prewashed and rinsed with deionized water containing 20% HNO3 [62]. The bottles containing the surface water samples were then pretreated with a few drops of concentrate ($HNO_3$, 65%) to prevent microbial degradation of heavy metals [24]. The bottles were capped, labelled, and shipped to the Botany Department Laboratory, Islamia College Peshawar, and stored at 4˚C until further analysis.

## Analytical methods

Physiochemical properties such as pH, temperature, electrical conductivity (EC), and total dissolved solids (TDS) were measured using a pH meter (Model PH-2012 China) and TDS/EC meter (Model-WT01 China) using standard operating protocols [63]. Standard titration methods were used for the measurement of chlorides ($Cl^-$) and $CaCO_3$ by titrating the sample with $H_2SO_4$, EDTA, and $AgNO_3$ [64]. The sulfate ($SO_4^{2-}$) concentration was determined by UV spectrophotometry (HACH-2800) [65]. Biological oxygen demand (BOD) and chemical oxygen demand (COD) were measured using standard methods [63].

Analytical grade chemicals ($HNO_3$, $H_2SO_4$) were used for acid digestion of the collected samples. First, 5 mL of concentrated $HNO_3$ was added to 250 mL Erlenmeyer flasks containing 100 mL of a contaminated surface water sample. The flasks were covered with a ribbed watch glass and boiled slowly on a hot plate, and the sample volume was reduced to 20 mL [63]. After cooling, concentrated $HNO_3$ and $H_2SO_4$ were further added to the flasks at a ratio of 5 to 10 mL and evaporated on a hot plate until dense white fumes appeared. All samples were cooled, filtered with Whatman no. 0.45 μm filter paper, and diluted to 50 ml with distilled water [66]. Concentrations of aluminum (Al), calcium (Ca), chromium (Cr), copper (Cu), iron (Fe), magnesium (Mg), potassium (K), manganese (Mn), sodium (Na), nickel (Ni), phosphorous (P), silicon (Si), strontium (Sr), titanium (Ti), and zinc (Zn) were measured using ICP–OES spectrometer (Thermo Scientific, iCAP 6000, UK) according to the measurement protocols of [67, 68].

## Quality control and quality assurance

Special attention was given to the collection, preservation, and each experimental technique of the samples. The samples were analyzed according to the standard certified analytical quality control method [69–72]. All analytical reagents used in the laboratory analysis were pure. The standard reference material (SRM) used for the preparation of calibration curves was Multielement Plasma Standard Solution 4, Refractory Metals Plasma Standard, and Phosphorus Atomic Absorption Standard produced by Thermo Fisher Scientific, USA. Seven different linear concentration standards were prepared. All metals with a good linear graph with a correlation coefficient > 0.999 were observed in the preparation of standard curves. Before sample analysis, groundwater, and wastewater pollution standard solution 1 and 5, Specpure (Thermo Fisher Scientific, USA) were run immediately after calibration. The recoveries were within the EPA limits of 95–105% (S1 Table). The accuracy of the ICP–OES analytical technique was verified by analyzing certified reference materials (CRMs) NIST SRM 1643F and 2781 (S1 Table). The limit of detection (LOD) for each element was determined by digesting five blanks (S1 Table). Each sample was analyzed in triplicate, and its mean was used for result interpretation. The relative standard deviation (RSD) values of the elements analyzed were ≤ 2.0% (S1 Table).

## Assessment of surface water quality for irrigation

Water quality assessment was primarily developed to determine the mineral composition of water [73]. In the current study, the suitability of surface water for irrigation purposes was evaluated using the sodium adsorption ratio (SAR) in conjunction with the sodium percentage

(Na%), Kelly's ratio (KR), and magnesium adsorption ratio (MAR) by using Eqs (1–4) [38, 74–77].

$$SAR = \frac{Na^+}{\frac{(\sqrt{Ca^{2+} + Mg^{2+}})}{2}} \tag{1}$$

$$Na\ (\%) = \frac{Na^+}{(Na^+ + K^+ + Ca^{2+})} X\ 100 \tag{2}$$

$$MAR = \frac{Mg^{2+}\ X\ 100}{Ca^{2+} + Mg^{2+}} \tag{3}$$

$$KR = \frac{Na^+}{Ca^{2+} + Mg^+} \tag{4}$$

The water quality and suitability for irrigation according to the above criteria are defined in S2 Table.

## Health risk assessment

Human health risk assessment is an approach commonly used to calculate the nature and likelihood of adverse health effects on humans [78]. Workers in the marble industry and farmers in agricultural fields are more exposed to contaminated surface water through ingestion and dermal pathways during irrigation and stone crushing [28, 79]. In addition, surface water with marble runoff intrudes into shallow aquifers and contaminates freshwater resources [23]. Consequently, assessment of the carcinogenic and noncarcinogenic risks to individual health from exposure to heavy metal-contaminated surface water via ingestion and dermal routes is of paramount importance. Therefore, metal concentrations in surface water were used to calculate human health risks through ingestion and dermal exposure routes. The adult and child population groups were considered based on their exposure frequency [28, 79].

## Noncarcinogenic health risk

The potential noncarcinogenic health risks from exposure to contaminated surface water via ingestion and dermal route were calculated according to the guidelines of [80] using Eqs 5 and 6.

$$ADD_{ing} = \frac{C_w\ x\ IR\ x\ EF\ x\ ED}{BW\ x\ AT} \tag{5}$$

$$ADD_{derm} = \frac{(C_w\ x\ SA\ x\ K_p\ x\ EF\ x\ ET\ x\ ED\ x\ CF)}{BW\ x\ AT} \tag{6}$$

where $ADD_{ing}$, and $ADD_{derm}$ is the average daily absorbed dose through ingestion and dermal contact [81]. The input parameters of Eqs 5 and 6 with relevant values are presented in S3 Table.

The hazard quotient (HQ) is a measure of potential noncarcinogenic health risk estimated by comparing the daily average absorbed dose (ADD) from the ingestion and dermal routes of exposure to the respective reference dose (RfD) by the following relationship [82].

$$HQ_{ing} = \frac{ADD_{ing}}{RfD_{ing}} \tag{7}$$

$$HQ_{derm} = \frac{ADD_{derm}}{RfD_{derm}} \tag{8}$$

where $HQ_{ing}$ and $HQ_{derm}$ refer to the hazard quotient from ingestion and dermal absorption route and $AAD_{ing}$ and $ADD_{derm}$ are the average daily absorbed doses derived from Eqs 5 and 6. $RfD_{ing}$ and $RfD_{derm}$ are ingestion and dermal reference doses and can be defined as the maximum tolerable risks for the human population while maintaining a vulnerable group throughout life [83]. The oral reference doses for Al, Cr, Cu, Fe, Mn, Ni and Zn were 1.0, 0.003, 0.04, 0.7, 0.02, 0.02 and 0.3 mg/kg/day; the dermal reference doses were 0.2, 0.000075, 0.012, 0.045, 0.0008, 0.0054 and 0.06 mg/kg/day [80, 84–87]. The noncarcinogenic health risk was neglected at HQ < 1.0 and taken into account at HQ > 1.0 [78].

The hazard index (HI) measures the total noncarcinogenic risk posed by multiple heavy metals present in the sample and is the sum of the hazard quotients of all heavy metals [82]. The HI can be calculated by the following equations:

$$HI_{ing} = \sum_{i=1}^{n} HQ_{ing=HQ_{Al}+HQ_{Cr}+HQ_{Cu}+HQ_{Fe}+HQ_{Mn}+HQ_{Ni}+HQ_{Zn}} \tag{9}$$

$$HI_{derm} = \sum_{i=1}^{n} HQ_{derm=HQ_{Al}+HQ_{Cr}+HQ_{Cu}+HQ_{Fe}+HQ_{Mn}+HQ_{Ni}+HQ_{Zn}} \tag{10}$$

$$HI = HI_{ing} + HI_{derm} \tag{11}$$

HI, values greater than one (HI > 1) indicate a potential for adverse human health effects or the need for additional studies [80].

## Carcinogenic risk assessment

The carcinogenic risk from exposure to the potentially toxic elements was estimated by ingestion and dermal route using incremental lifetime cancer risk (ILRC) [88]. ILCR is the incremental probability that a person will develop a type of cancer over their lifetime as a result of exposure to a carcinogenic hazard [84]. Cancer risk can be calculated using the following equations developed by [80].

$$ILCR = ADD \text{ x } CSF_{ABS} \tag{12}$$

$$SF_{ABS} = \frac{CSF_0}{ABS_{GI}} \tag{13}$$

where ADD is the average daily dose (mg/kg/day), $CSF_{ABS}$ is the absorbed cancer slope factor, and $CSF_O$ is the oral cancer slope factor, which represents the risk posed by the lifetime average amount of one mg/kg/day of a chemical carcinogen and toxin specific [11]. $ABS_{GI}$ is the absorption fraction in the gastrointestinal tract (USEPA, 2004). Among the elements examined, Cr and Ni were identified as potentially carcinogenic to humans and included in the list of the International Agency for Research on Cancer [86]. The oral slope factors ($CSF_O$) and gastrointestinal absorption fraction ($ABS_{GI}$) were 0.5 mg/kg/day and 2.5% for Cr and 0.84 mg/kg/day and 4% for Ni [80, 86, 89]. An ILCR value above $1 \times 10^{-6}$ is considered harmful, while a value below $1 \times 10^{-6}$ is accepted as a nonsignificant risk [28].

**Table 1. Mean concentrations of physicochemical parameters of water (n = 27) used for surface irrigation near the marble industry in District Malakand.**

| Parameters | Sampling sites | | | | | | | | | Statistics | | | | Standard | | | |
|---|---|---|---|---|---|---|---|---|---|---|---|---|---|---|---|---|---|
| | C | S1 | S2 | S3 | S4 | S5 | S6 | S7 | S8 | Mean | Min | Max | SD | NEQs | WHO | USEPA | Increase (%) |
| pH | 7.4 | 8.1 | 7.8 | 7.7 | 7.9 | 8.2 | 8.1 | 8.1 | 8.3 | 7.96 | 7.4 | 8.3 | 0.28 | 6–9 | 6.5–8.5 | 6–8.4 | - |
| EC µs/cm | 175.7 | 472.3 | 467.7 | 585.3 | 638.3 | 557.3 | 443.3 | 323.0 | 274.7 | 437.51 | 175.7 | 638.3 | 152.82 | 400 | 400 | 700 | 67% |
| TDS mg/L | 112.0 | 302.0 | 299.0 | 375.0 | 409 | 357.0 | 284.0 | 207.0 | 176.0 | 280.11 | 112.0 | 409.0 | 98.00 | 3500 | 450 | - | - |
| Tem C˚ | 23.6 | 27.7 | 28.0 | 31.2 | 31.8 | 27.5 | 31.3 | 29.7 | 29.0 | 28.87 | 23.6 | 31.8 | 2.56 | 40 | - | - | - |
| BOD mg/L | 74.0 | 181.7 | 197.0 | 182.0 | 91.3 | 102.0 | 168.3 | 98.7 | 184.3 | 142.14 | 74.0 | 197.0 | 49.18 | 80 | - | 30 | 89 |
| COD mg/L | 98.3 | 174.7 | 212.7 | 172.0 | 209.7 | 182.3 | 174.0 | 248.7 | 232.0 | 189.38 | 98.3 | 248.7 | 43.86 | 150 | - | 120 | 89 |
| Cl⁻ mg/L | 36.8 | 44.6 | 86.6 | 46.9 | 55.1 | 61.9 | 86.0 | 70.2 | 69.8 | 61.99 | 36.8 | 86.6 | 17.76 | 1000 | 105 | 98 | - |
| SO₄²⁻ mg/L | 90.0 | 407.3 | 188.3 | 295.0 | 380.7 | 462.3 | 229.0 | 270.3 | 455.0 | 308.66 | 90.0 | 462.3 | 127.59 | 600 | - | - | - |
| CaCO₃ mg/L | 60.7 | 126.3 | 122.0 | 86.0 | 120.0 | 195.0 | 114.0 | 134.3 | 122.0 | 120.03 | 60.7 | 195.0 | 36.38 | - | 500 | - | - |
| Ca²⁺ mg/L | 76.5 | 521.0 | 378.0 | 205.0 | 331.0 | 185.2 | 455.0 | 852.0 | 112.0 | 346.19 | 76.5 | 852.0 | 243.13 | 100 | - | - | 89 |
| Mg²⁺ mg/L | 8.75 | 413.0 | 113.0 | 32.7 | 92.0 | 36.9 | 187.0 | 37.0 | 15.7 | 104.01 | 8.75 | 413.0 | 129.20 | 150 | - | - | 23 |
| Na⁺ mg/L | 131.8 | 301.5 | 278.0 | 220.3 | 334.0 | 236.6 | 264.7 | 410.3 | 376.3 | 283.72 | 131.8 | 410.3 | 84.52 | 200 | - | - | 89 |
| K⁺ mg/L | 8.31 | 9.55 | 51.0 | 9.89 | 46.1 | 19.6 | 11.5 | 7.3 | 7.5 | 18.97 | 7.3 | 51.0 | 17.21 | 12 | - | - | 34 |

SD = Standard deviation, NEQs = National environmental Quality standard, EPA = Environmental protection agency, WHO = World Health organization

## Statistical analysis

Descriptive statistical analysis were performed for the investigated physicochemical parameters and heavy metals with related health risks using SPSS statistical package (version 25) and Excel 2016 software (Tables 1–5). Compositional data analysis (CoDa) techniques, such as centered log-ratio transformation (CLR), were applied to raw data. A log ratio-variation matrix (Table 6), CLR covariance biplots (Fig 5), CoDa-PCA (Table 7) and a CoDa-dendrogram (Fig 4) were constructed to find the linear relationship between elements and source attribution [50–52]. For these calculations, the CoDaPack software [90] and R package 'compositions' [91] were used. The spatial distribution of elements in water samples near the marble industry was obtained by inverse distance-weighted interpolation (IDW) in ArcGIS version 10.2.2 [92].

**Table 2. Classification of the irrigation water quality of the study area.**

| Site | SAR (meq/L) | | EC µs/cm | | MAR (meq/L) | | KR (meq/L) | | Na% (meq/L) | |
|---|---|---|---|---|---|---|---|---|---|---|
| | Value | Description | Value | Description | Value | Description | Value | Description | Value | Description |
| C | 3.81 | Excellent | 175.70 | Excellent | 15.89 | Suitable | 0.79 | Suitable | 56.44 | Permissible |
| S1 | 2.60 | Excellent | 472.30 | Good | 53.47 | Unsuitable | 34.57 | Unsuitable | 20.59 | Good |
| S2 | 3.22 | Excellent | 467.70 | Good | 32.95 | Suitable | 9.91 | Unsuitable | 32.24 | Permissible |
| S3 | 3.86 | Excellent | 585.30 | Good | 22.23 | Suitable | 3.73 | Unsuitable | 44.17 | Permissible |
| S4 | 4.19 | Excellent | 638.30 | Good | 31.56 | Suitable | 8.45 | Unsuitable | 39.42 | Permissible |
| S5 | 4.15 | Excellent | 557.30 | Good | 24.67 | Suitable | 4.16 | Unsuitable | 46.61 | Permissible |
| S6 | 2.63 | Excellent | 443.30 | Good | 40.47 | Suitable | 15.93 | Unsuitable | 23.44 | Good |
| S7 | 3.74 | Excellent | 553.70 | Good | 6.65 | Suitable | 3.45 | Unsuitable | 28.33 | Good |
| S8 | 8.92 | Excellent | 274.70 | Good | 19.06 | Suitable | 4.33 | Unsuitable | 70.23 | Unsuitable |
| Mean | 4.13 | - | 463.14 | - | 27.44 | | 9.64 | - | 40.16 | |
| Min | 2.60 | - | 175.70 | - | 6.65 | | 0.79 | - | 20.59 | |
| Max | 8.92 | - | 638.30 | - | 53.47 | | 34.57 | - | 70.23 | |
| SD | 1.9 | - | 150.54 | - | 14.0 | | 10.3 | - | 16.2 | |

SAR = Sodium adsorption ratio, MAR = Magnesium adsorption ratio, KR = Kelly's ratio, EC = Electrical conductivity

**Table 3. Heavy metal load (mg/L) in contaminated surface water (n = 27) used for irrigation of food crops.**

| Elements | Sampling sites | | | | | | | | | Statistics | | | | Standard | |
|---|---|---|---|---|---|---|---|---|---|---|---|---|---|---|---|
| | C | S1 | S2 | S3 | S4 | S5 | S6 | S7 | S8 | Mean | Min | Max | SD | FAO/NEQS | Increase (%) |
| Al | 3.13 | 7.14 | 565 | 129 | 469 | 10.9 | 9.39 | 5.05 | 7.79 | 134.04 | 3.13 | 565 | 222.06 | 5.0 | 89 |
| Cr | 0.11 | 0.07 | 0.31 | 0.07 | 0.54 | 0.11 | 0.11 | 0.14 | 0.09 | 0.17 | 0.07 | 0.54 | 0.16 | 0.1 | 56 |
| Cu | 0.12 | 0.28 | 1.27 | 0.23 | 0.97 | 0.20 | 0.39 | 0.67 | 1.07 | 0.58 | 0.12 | 1.27 | 0.43 | 0.2 | 78 |
| Fe | 1.97 | 2.93 | 93.6 | 15.7 | 113 | 5.0 | 7.15 | 21.5 | 2.96 | 29.31 | 1.97 | 113 | 42.72 | 5.0 | 56 |
| Mn | 0.09 | 0.15 | 1.57 | 0.2 | 0.72 | 0.14 | 0.39 | 0.54 | 0.29 | 0.45 | 0.09 | 1.57 | 0.47 | 0.2 | 56 |
| Ni | 0.17 | 0.31 | 0.30 | 0.09 | 0.27 | 0.09 | 0.43 | 0.65 | 0.11 | 0.27 | 0.09 | 0.65 | 0.18 | 0.2 | 56 |
| P | 1.79 | 12.8 | 12.3 | 14.4 | 12.1 | 26.5 | 11.9 | 11.7 | 30.5 | 14.89 | 1.79 | 30.5 | 8.58 | 5 | 89 |
| Si | 4.36 | 2.29 | 1.20 | 2.31 | 1.41 | 4.46 | 3.71 | 7.09 | 1.29 | 3.12 | 1.2 | 7.09 | 1.96 | - | - |
| Sr | 0.25 | 0.35 | 2.29 | 0.65 | 2.18 | 0.95 | 1.79 | 1.36 | 0.41 | 1.14 | 0.25 | 2.29 | 0.8 | - | - |
| Ti | 0.04 | 1.1 | 10.13 | 1.41 | 12.9 | 0.31 | 0.11 | 0.09 | 0.14 | 2.91 | 0.04 | 12.9 | 4.95 | - | - |
| Zn | 1.78 | 2.26 | 1.51 | 1.49 | 1.47 | 1.79 | 1.71 | 1.45 | 1.49 | 1.66 | 1.45 | 2.26 | 0.26 | 2.0 | 11 |

SD = Standard deviation, NEQS = National environmental Quality standard, FAO = Food and agricultural organization

# Results and discussion

## Assessment of surface water quality for irrigation

The data for evaluating surface water quality for irrigation purposes near marble industry was analyzed statistically and their results are summarized in Tables 1 and 2, respectively. The pH values ranged from 7.4 to 8.3, with a mean of 7.96, indicating a slightly alkaline nature (Table 1). All water samples were within the standard safe pH limit (6.5–8.5) for irrigation purposes [93, 94]. pH is a term commonly used to express the intensity of a solution's acidic or alkaline state [95]. pH values above 6.5 to 8.5 are suitable for irrigation but lead to nutrient

**Table 4. Non-carcinogenic risk assessment for children and adults from heavy metal exposure via ingestion and dermal pathways.**

| Elements | ADD ingestion | ADD dermal | | HQ ingestion | HQ dermal | HQ total |
|---|---|---|---|---|---|---|
| **Adults** | | | | | | |
| Al | 3.67E+00 | 1.92E-02 | | 3.67E+00 | 9.58E-02 | 3.77E+00 |
| Cr | 4.72E-03 | 4.92E-05 | | 1.57E+00 | 6.57E-01 | 2.23E+00 |
| Cu | 1.58E-02 | 8.27E-05 | | 3.96E-01 | 6.89E-03 | 4.03E-01 |
| Fe | 8.03E-01 | 4.20E-03 | | 1.15E+00 | 9.31E-02 | 1.24E+00 |
| Mn | 1.25E-02 | 6.51E-05 | | 6.23E-01 | 8.13E-02 | 7.04E-01 |
| Ni | 7.37E-03 | 1.54E-04 | | 3.68E-01 | 2.85E-02 | 3.97E-01 |
| Zn | 4.55E-02 | 1.43E-04 | | 1.52E-01 | 2.38E-03 | 1.54E-01 |
| Total | 4.56E+00 | 2.39E-02 | **HI** | 7.93E+00 | 9.65E-01 | 8.90E+00 |
| **Child** | | | | | | |
| Al | 5.48E+00 | 5.65E-02 | | 5.48E+00 | 2.83E-01 | 5.77E+00 |
| Cr | 7.05E-03 | 1.45E-04 | | 2.35E+00 | 1.94E+00 | 4.29E+00 |
| Cu | 2.36E-02 | 2.44E-04 | | 5.91E-01 | 2.03E-02 | 6.11E-01 |
| Fe | 1.20E+00 | 1.24E-02 | | 1.71E+00 | 2.75E-01 | 1.99E+00 |
| Mn | 1.86E-02 | 1.92E-04 | | 9.30E-01 | 2.40E-01 | 1.17E+00 |
| Ni | 1.10E-02 | 4.54E-04 | | 5.50E-01 | 8.40E-02 | 6.34E-01 |
| Zn | 6.80E-02 | 4.20E-04 | | 2.27E-01 | 7.01E-03 | 2.34E-01 |
| Total | 6.81E+00 | 7.04E-02 | **HI** | 1.18E+01 | 2.85E+00 | 1.47E+01 |

**Table 5. Carcinogenic risk (CR) of Cr and Ni for adult and children through ingestion and dermal exposures pathways.**

| Adult | | | | Children | | | |
|---|---|---|---|---|---|---|---|
| Element | $CR_{ingestion}$ | $CR_{dermal}$ | $CR_{total}$ | $CR_{ingestion}$ | $CR_{dermal}$ | $CR_{total}$ | |
| Cr | $2.4 \times 10^{-3}$ | $2.5 \times 10^{-5}$ | $2.4 \times 10^{-3}$ | $3.5 \times 10^{-3}$ | $7.3 \times 10^{-5}$ | $3.6 \times 10^{-3}$ | |
| Ni | $6.6 \times 10^{-3}$ | $1.4 \times 10^{-4}$ | $6.8 \times 10^{-3}$ | $9.9 \times 10^{-3}$ | $4.1 \times 10^{-4}$ | $1.0 \times 10^{-2}$ | |
| Total | $9.0 \times 10^{-3}$ | $1.6 \times 10^{-4}$ | $9.2 \times 10^{-3}$ | $1.34 \times 10^{-2}$ | $5 \times 10^{-4}$ | $1.39 \times 10^{-2}$ | |

imbalances and alter the solubility of many toxic substances [96]. Temperature values ranged from 23.60 to 31.80˚C with a mean of 28.87˚C and are within the safe limit (40˚C) for irrigation [93]. According to [97], an increase in water temperature decreases dissolved oxygen (DO) solubility and has a strong impact on nutrient recycling and the productivity of aquatic biodiversity.

The BOD of wastewater is a valuable criterion for assessing its suitability for irrigation [98]. High BOD in wastewater depletes the oxygen content of the water and leads to the death of aquatic organisms [99]. Similar to BOD, COD is also used to represent overall water quality [100]. The BOD and COD concentrations in the current study ranged from 74 to 197 mg/L and 98.30 to 248.70 mg/L, with mean values of 142.14 and 189.38 mg/L, respectively (Table 1). The permissible limits for BOD and COD in wastewater are 80 and 150 mg/L, respectively [93]. The concentrations of BOD and COD in the water sample (89%) exceeded the safety limits for irrigation (Table 1). High BOD and COD levels were also obtained by [64] when studying the impact of marble runoff on the Barandu River in Pakistan. Alkalinity is essential to aquatic life due to its buffering capacity against rapid pH changes that occur naturally as a result of the photosynthetic activity of plants [101]. In the current study, alkalinity ranged from 60.7 to 195 mg/L with a mean of 120.03 mg/L CaCO3 (Table 1). The alkalinity levels of all water samples were within the safe limit (500 mg/L) for irrigation purposes [93].

Concentrations of the cations including $Ca^{2+}$, $Na^+$, $Mg^{2+}$, and $K^+$ ranged from 76.5 to 852, 131.8 to 410.3, 8.75 to 413, and 7.3 to 51 mg/L with mean values of 346.19, 283.72, 104.01 and 18.97 mg/L, respectively (Table 1). The permissible limits of $Ca^{2+}$, $Na^+$, $Mg^{2+}$, and $K^+$ in wastewater used for irrigation are 100, 200, 150, and 12 mg/L, respectively [93]. Due to these

**Table 6. Normalized variation matrix of data in table (3).**

| | Al | Ca | Cr | Cu | Fe | K | Mg | Mn | Na | Ni | p | Si | Sr | Ti | Zn |
|---|---|---|---|---|---|---|---|---|---|---|---|---|---|---|---|
| Al | | | | | | | | | | | | | | | |
| Ca | 2.12 | | | | | | | | | | | | | | |
| Cr | 1.31 | 0.50 | | | | | | | | | | | | | |
| Cu | 1.64 | 0.50 | 0.3 | | | | | | | | | | | | |
| Fe | 0.67 | 1.36 | 0.73 | 1.04 | | | | | | | | | | | |
| K | 1.09 | 0.52 | **0.17** | 0.39 | 0.95 | | | | | | | | | | |
| Mg | 2.12 | 0.34 | 0.89 | 0.88 | 1.95 | 0.71 | | | | | | | | | |
| Mn | 1.39 | 0.50 | 0.37 | **0.20** | 0.84 | 0.47 | 0.87 | | | | | | | | |
| Na | 2.04 | 0.21 | 0.36 | 0.42 | 1.42 | 0.41 | 0.65 | 0.54 | | | | | | | |
| Ni | 2.47 | **0.19** | 0.44 | 0.60 | 1.48 | 0.63 | 0.58 | 0.56 | 0.3 | | | | | | |
| P | 2.69 | 0.82 | 0.77 | 0.80 | 2.13 | 0.94 | 1.32 | 0.90 | 0.46 | 0.57 | | | | | |
| Si | 3.26 | 0.49 | 0.74 | 0.98 | 2.12 | 0.86 | 1.13 | 1.16 | 0.38 | 0.4 | 0.78 | | | | |
| Sr | 1.40 | 0.36 | 0.25 | 0.41 | 0.79 | 0.31 | 0.75 | 0.46 | 0.44 | 0.47 | 1.03 | 0.79 | | | |
| Ti | 0.49 | 2.19 | 1.47 | 1.71 | 1.08 | 1.23 | 1.92 | 1.63 | 2.24 | 2.56 | 3.0 | 3.46 | 1.56 | | |
| Zn | 2.25 | 0.36 | 0.40 | 0.59 | 1.63 | 0.43 | 0.73 | 0.74 | **0.13** | 0.33 | 0.43 | 0.23 | 0.55 | 2.37 | |

**Table 7. Principal component analysis (PCA) based on centered log ratio (CLR) transformed data.**

| CLR (variables) | Components | |
|---|---|---|
| | PC1 | PC2 |
| Al | **0.52** | -0.05 |
| Ca | -0.16 | **0.22** |
| Cr | 0.01 | -0.19 |
| Cu | -0.02 | -0.13 |
| Fe | **0.32** | -0.43 |
| K | 0.05 | 0.05 |
| Mg | -0.09 | **0.72** |
| Mn | 0.03 | -0.13 |
| Na | -0.17 | -0.01 |
| Ni | -0.22 | 0.04 |
| P | -0.25 | -0.25 |
| Si | -0.35 | -0.10 |
| Sr | 0.01 | -0.04 |
| Ti | **0.53** | 0.32 |
| Zn | -0.21 | -0.01 |
| Eigen | 14.17 | 2.41 |
| % Variance | 64.00 | 11.00 |
| Total variance | 64.00 | 75.00 |

permissible limit values, the percentages (%) of water samples with $Ca^{2+}$ (89), $Na^+$ (89), $Mg^{2+}$ (23) and $K^+$ (34) are unsuitable for irrigation (Table 1). Likewise, the concentrations of anions such as $SO_4^{2-}$ and $Cl^-$ were between 90 and 462.3, 36.8, and 86.6 mg/L with mean values of 308.66 and 61.99 mg/L, respectively (Table 1). The permissible limits set by [93] for $SO_4^{2-}$ and $Cl^-$ in wastewater used for irrigation are 600 and 1000 mg/L, respectively (Table 1). According to the classification standards, all the water samples are suitable for irrigation in terms of $SO_4^{2-}$ and $Cl^-$ (Table 1).

Investigating salinity risk is very important for irrigation water, as a high salt content renders the soil saline and affects the ability of plants to absorb water through their roots [102]. Electrical conductivity (EC) is a measure of water's ability to conduct electricity and indicates the number of total dissolved solids (TDS) [103]. Therefore, the salinity hazard of the current study was assessed by EC and TDS, and their concentrations varied from 175.7 to 638.3 μs/cm and 112 to 409 mg/L with mean values of 437.51 μs/cm and 280.11 mg/L, respectively (Table 1). Based on the EC level, 67% of the surface water samples were above the permissible limits (400 μs/cm) set by [93, 94] for irrigation. However, the TDS concentration in all samples is within the permissible limit (450–3500 Mg/L) set by [93, 94] for wastewater (Table 1). In addition, the long-term use of this slightly saline water for irrigation may increase the salinity risk in the soils of the study area.

An assessment of the sodium hazard of water is required to determine its suitability for irrigation. Excessive sodium levels in the water sample reduce the permeability and availability of water for the plant [104]. One of the most important criteria in determining sodium hazard is the sodium adsorption ratio (SAR). In addition, Kelly's ratio (KR) and Na% are used to assess the adverse effect of sodium (Na) on irrigation water quality [105]. The SAR ranged from 2.60 to 8.92 meq/L, with a mean of 4.13 (Table 2). In the current study, SAR values < 10 classify the water samples as good for irrigation (S2 Table). However, the Kelly's ratio (KR > 1) indicates excess sodium in water. Water with Kelly's ratios (K < 1) is considered suitable for irrigation

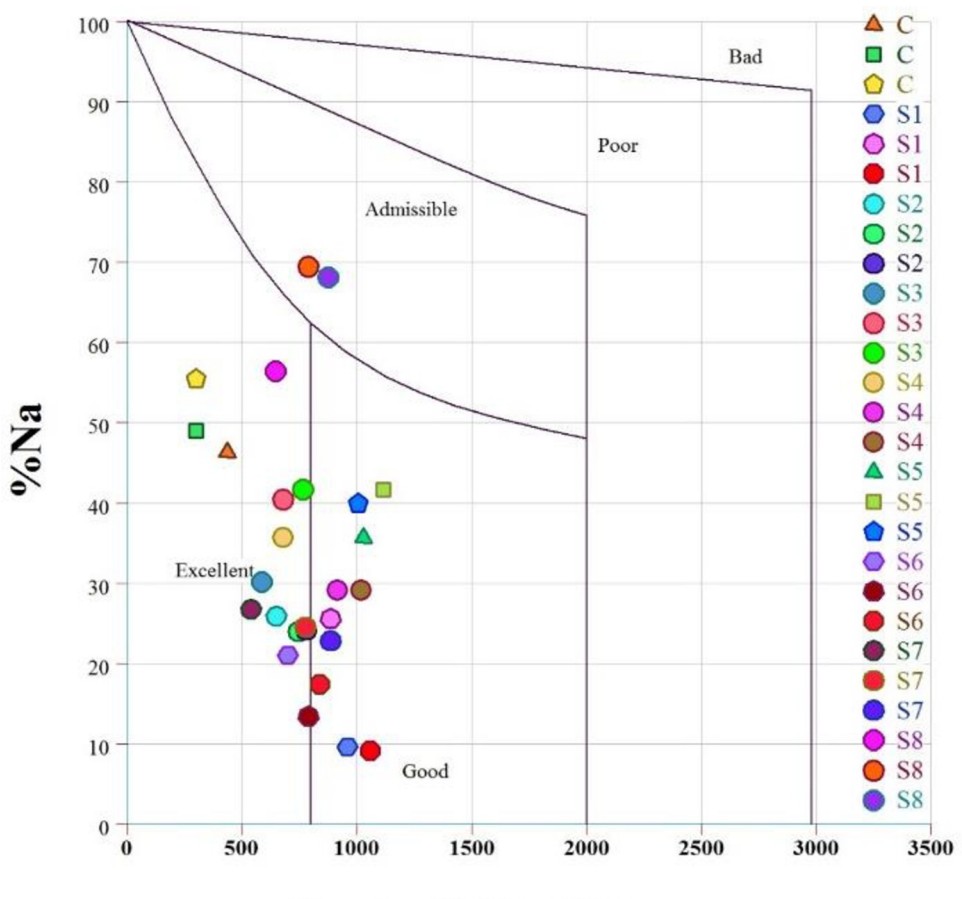

**Fig 3. The Wilcox diagram shows the suitability of surface water for irrigation.**

while K > 1 is considered as unsuitable [95]. Kelly's ratios in the present study ranged from 0.79 to 34.57 meq/L, with a mean of 9.64 meq/L (Table 2). Based on these values, 89% of the water samples were classified as unsuitable for irrigation (K > 1) (Table 2). The Na% ranged from 20.59 to 70.23 meq/L, with a mean of 40.16 meq/L (Table 2). Na% is plotted against conductivity, called the Wilcox diagrams, shown in Fig 3. The water sample (52%) falls into the categories excellent, 40% good and 8% admissible. The magnesium adsorption ratio (MAR) values ranged from 6.65 to 53.47, with a mean of 27.44 meq/L (Table 2). Based on MAR, 88% of the water sample was suitable for irrigation while the remaining 12% was considered as unsuitable (Table 2). In general, calcium and magnesium maintain a state of equilibrium in most water bodies, while increased concentrations of magnesium in the water reduce crop yield [106].

## Heavy metal (HM) concentration in surface water

The HM concentrations in surface water, including marble runoff, are summarized in Table 3. The concentrations of Al, Fe, P, Ti, and Si ranged from 3.13 to 565.0, 1.97 to 113, 1.79 to 30.5, 0.04 to 12.90, and 1.20 to 7.09 mg/L with mean values of 134.04, 29.31, 14.89, 2.91 and 3.12 mg/L, respectively. Likewise, the concentrations of Zn, Sr, Cu, Mn, Ni, and Cr range from 1.45 to 2.26, 0.25 to 2.29, 0.12 to 1.27, 0.09 to 1.57, 0.09 to 0.65, and 0.07 to 0.54 mg/L, with mean

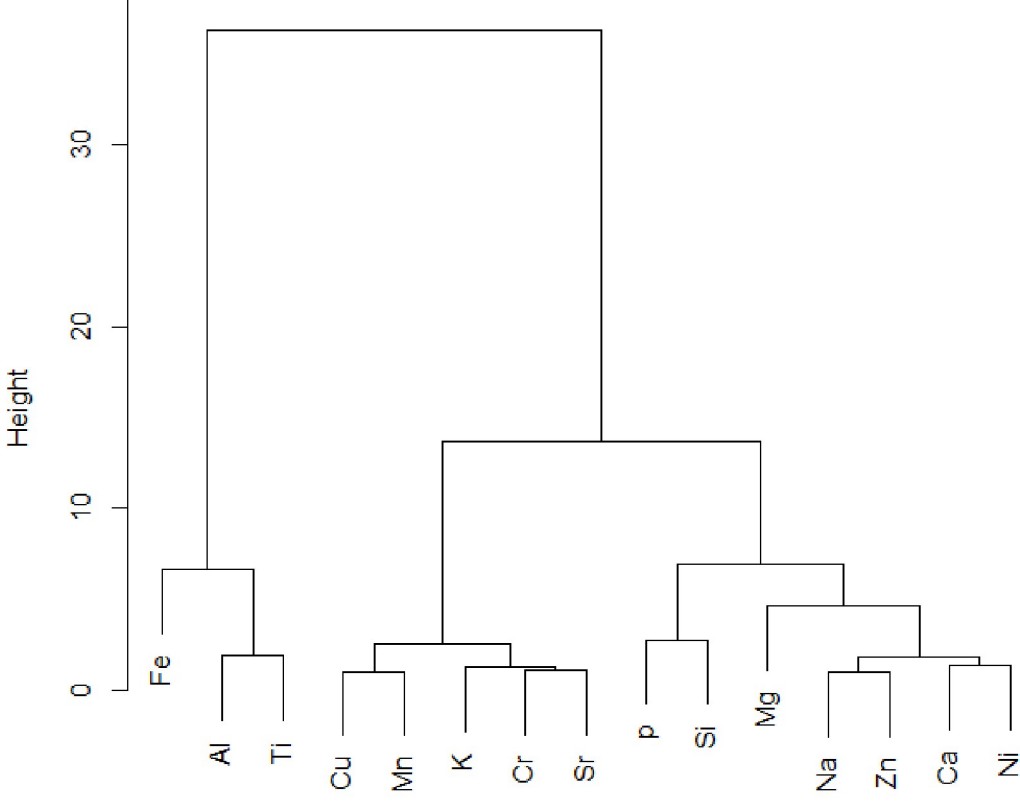

**Fig 4. Dendrogram of the association of the fifteen elements in the water sample reproduced by the R-package 'compositions' and 'hclust.**

values of 1.66, 1.14, 0.58, 0.45, 0.27 and 0.17 mg/L, respectively. The permissible limits for Al, Cr, Cu, Fe, Mn, Ni, P, and Zn in wastewater were 5.0, 0.1, 0.2, 5.0, 0.2, 0.2, 5.0, and 11 mg/L, respectively (Table 3). The proportions of Al (89%), Cu (78%), Fe (56%), Mn (56%), Ni (56%), Cr (56%), and Zn (11%) were above the safety limits for irrigation [93]. Due to unavailability of threshold values for Si, Sr, and Ti in wastewater. Consequently, their assessment for irrigation and health risk was not reported in the present study. According to [107] elevated Al concentrations in water are toxic to gill-breathing animals and lead to loss of their osmoregulatory function. Likewise, increased levels of Cr, Fe, and Pb lead to severe histopathological changes in fish liver, gills, and kidneys [108, 109]. Concentrations of Fe and Zn above the permissible limit in aquatic ecosystems reduce aquatic biodiversity and cause necrosis in the flora [110, 111]. However, excessive Fe concentrations in the human body lead to gastrointestinal tract disorders and an unnecessary rise in blood pressure [112]. Additionally, Cu, Ni, and Zn above safe limits damage the liver, kidneys, and pancreas and cause nausea, fever, and skin problems [113, 114]. Elevated Mn concentrations in water are associated with mining and industrial effluents, fossil fuel burning, and the steel and marble industries [23, 115]. Prolonged exposure to Mn can target the mitochondria and cause neurotoxicity, followed by liver cirrhosis in

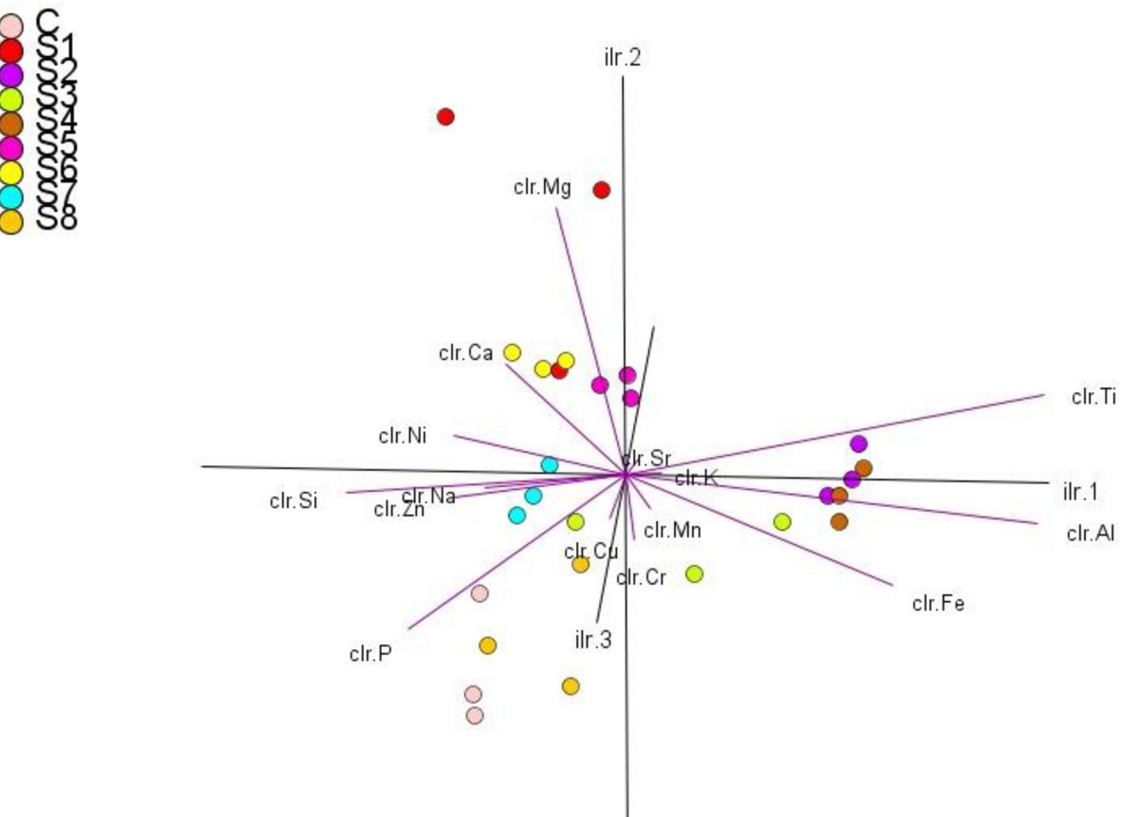

**Fig 5. Centered log ratio (CLR) transformation covariance biplots shows clustering of elements on PC1 versus PC2.**

humans [116]. Similarly, Ni levels above the safe limit can cause lung, throat, and stomach cancers in humans [117].

### Health risk assessment

**Noncarcinogenic risk assessment.** The average daily dose (ADD mg/kg/day), hazard quotient (HQ), and hazard index (HI) values of Al, Cr, Cu, Fe, Mn, Ni, and Zn were calculated and are presented in Table 4. The $ADD_{ing}$ values for Al (5.48E+00), Fe (1.20E+00), Zn (6.80E-02), Cu, (2.36E-02), Mn (1.86E-02), Ni (1.10E-02) and Cr (7.05E-03) were higher in children than in adults (Table 4). For adults, the $ADD_{ing}$ values of Al (3.67E+00), Fe (8.03E-01), Zn (4.55E-02), Cu (1.58E-02), Mn (1.25E-02), Ni (7.37E-02) and Cr (4.72E-03) were above the $ADD_{derm}$ calculated for children (Table 4). The $ADD_{derm}$ values for Al (5.65 E-02), Fe (1.24E-02), Ni (4.54E-04), Zn (4.20E-04), Cu (2.44E-04), Mn (1.92E-04), and Cr (1.45E-04) in children were higher than the estimated $ADD_{derm}$ for adults (Table 4). The trend of $ADD_{total}$ for both population through exposure routes showed a decreasing order of $ADD_{ing}$ (child) > $ADD_{ing}$ (adult) > $ADD_{derm}$ (child) > $ADD_{dem}$ (adult) (Table 4). Similarly, ingestion and dermal exposure to heavy metals were in decreasing order: Al > Fe > Ni > Zn > Cu > Mn > Cr. The current results from $ADD_{ing}$ and $ADD_{derm}$ revealed that children were more exposed to higher doses of heavy metals through ingestion and dermal routes than adults.

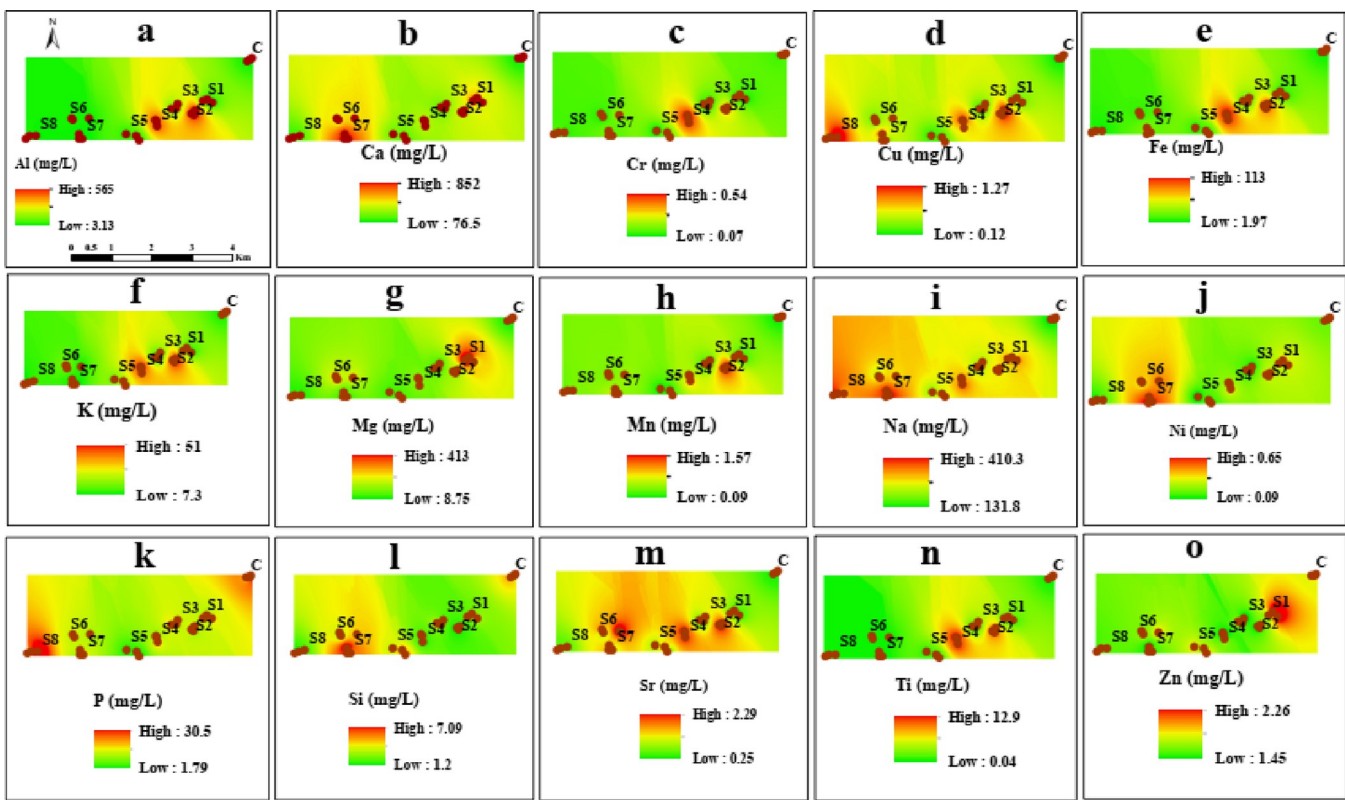

**Fig 6. Spatial distribution pattern of elements in irrigation water near the marble processing plants.**

The mean values of $HQ_{ing}$ ranged from 1.52E-01 to 3.67E+00 for adults and 2.38E-03 to 5.48E+00 for children (Table 4). HQ > 1.0 for Al, Cr, and Fe indicated adverse health effects in children and adults via the ingestion route of exposure (Table 4). However, $HQ_{derm}$ > 1.0 for Cr indicates potential health hazards for children via the dermal contact route. The risk of heavy metal exposure was in decreasing order of $HQ_{ing}$ (adult) > $HQ_{derm}$ (child) > $HQ_{ing}$ (child) > $HQ_{derm}$ (adult) (Table 4). The $HQ_{total}$ ($HQ_{ing}$ + $HQ_{derm}$) values for Al, Cr, Cu, and Fe were above one ($HQ_{total}$ > 1) (Table 4). Therefore, adverse health effects were suggested for the exposed population groups. In addition, the HI ($HQ_{ing}$ + $HQ_{derm}$) for the heavy metals in adults and children was also above one (HI > 1) (Table 4), indicating a significant risk of non-carcinogenic toxicity. The contribution from heavy metal (HM) to hazard index (HI) decreased in the order Al > Cr > Fe > Mn > Cu > Ni > Zn. Similar results were obtained by [28], who reported that irrigation with industrial wastewater is 180 times more hazardous than groundwater. Our current study results agree well with those of [38], who also calculated a noncarcinogenic risk to humans from heavy metal exposure via ingestion and dermal contact with contaminated surface water.

**Carcinogenic Risk Assessment (CRA).** The carcinogenic risk of Cr and Ni for children and adults through ingestion and dermal contact was calculated and ranged from $2.5\times10^{-5}$ to $9.9\times10^{-3}$ (Table 5). The minimum cancer risk value for Cr ($2.5\times10^{-5}$) from dermal exposure in adults exceeded the acceptable level of $1\times10^{-6}$–$1\times10^{-4}$ [118]. Likewise, the maximum cancer risk value of Ni ($9.9\times10^{-3}$) for children by the ingestion route is 10–1000 times higher than the [118] acceptable value of $1\times10-6$–$1\times10^{-4}$. The total cancer risk value of Ni and Cr was higher in children ($1.39\times10^{-2}$) than in adults ($9.2\times10^{-3}$)⸴ suggesting that children are more susceptible

to carcinogenic risk. According to [89], a carcinogenic risk of less than $1 \times 10^{-6}$ for individual heavy metals is insignificant and above $1 \times 10^{-4}$ can be quantified as harmful and of concern. Table 3 shows that higher concentrations of Cr and Ni in surface water, including marble run-off, pose a risk to human health. In addition, the cumulative cancer risk of the elements is more threatening than the individual element risks (Table 5).

## Looking for associations and identification of heavy metal sources in irrigation water

The normalized variation matrix suggests a linear association between sub compositional parts of a compositional data set [119]. Variations smaller than 0.2 indicate proportionality or linear association, and variations greater than 1.0 indicate a lack of linear association or proportionality [50]. In the current study, the smallest contribution from Na to Zn (0.13), Cr to K (0.17), Ca to Ni (0.19), and Cu to Mn (0.2) (Table 6), indicates that these pairs of elements are proportional and can easily be explained by the fact that sampling sites where Na is available are also rich in Zn, and the same is true for Cr and K, Ca and Ni, Cu and Mn. The largest contributor to variability is derived from the log ratios of Al, Ti, Fe, Mg, Si, and P relative to most elements (Table 6). The relatively large variation in the log ratios of Al, Ti, and Fe with Si, P, Ni, Zn, Na, Ca, Mg, and Cu (Table 6) separated Fe, Al, and Ti first in the cluster dendrogram (Fig 4). Disproportionate log ratios of Mg to Ca (0.34), Ni (0.58), Na (0.65), and Zn (0.73) removed Mg from their respective clusters (Fig 4). The associations of the remaining elements were Cu–Mn, K-Cr–Sr, P-Si, Na-Zn, and Ca-Ni (Fig 4). According to [120], metals falling into the same cluster lead to the same source of origin.

Principal component analysis (PCA) based on centered log ratio (CLR) transformation explains the inter elemental association and determines the pollution sources [121]. The result of the CLR biplot (Fig 5) represents 75% of the total variability (Table 7). The PCI was dominated by high loadings of Ti (0.53), Al (0.52), and Fe (0.32). PCII was dominated by high loadings of Mg (0.72), Ti (0.32), and Ca (0.22) (Table 7), indicating that these elements are from the same source. In Fig 5, the K, Ti, Al, and Fe variables form a clear association due to the same ray direction and the close layout of their vertices, implying that the association could have a combination of anthropogenic and geogenic origins [122]. Fig 5 also shows a close association between Na-Zn-Ni and Ca-Mg due to their close vertices and the same direction of the rays. Magnesium (Mg) and Si vectors are proportional to Na, Ca, Ni and Zn, indicating their dominance in carbonate rocks [123]. Copper (Cu), Mn, Cr, and Sr develop different associations as their vertices are positioned close together and the rays point in the same direction, possibly related to soil erosion, fertilizer, pesticides, and waste disposal from the municipal and industrial sectors [124, 125]. Silicon and phosphorus form a clear association due to the close location of their vertices, and the rays in the same direction are likely associated with soil erosion and phosphate fertilizer [123]. A similar study [64] reported that elements such as Al, Fe, Ca, K, and Ni are released in the form of dust from marble and granite rocks during mining activities, sculpting, and manufacturing of slabs. [53] also reported that the main anthropogenic sources of soil pollution from Cr, Pb, Mn, Ni, Cu, Mn, and Zn were the leather industry, brick factories, and road traffic and therefore strongly agree with our current study results.

## Spatial patterns of elements in surface water

The spatial distribution map is a valuable tool to identify hotspots of heavy metal pollution and delineate the safe and hazardous corners in a study area [69]. The spatial distribution of elements in surface water near the marble industry is shown in Fig 6. High concentration patterns of Al, K, Ti, and Fe were observed near the sampling sites (S2 to S5), indicating the

contribution of the marble industry to surface water pollution. Similarly, high concentration trends for Ca, Si, and Ni were recorded near sites S6 and S7 and could be a major contribution due to industrial and intensive agricultural practices along the water sources [126, 127]. An increase in the spatial distribution patterns of Cu, Na, P, and Sr was observed from control (C) to sites (S1 to S8), also indicating that marble runoff is an important source of surface water contamination. Cr, Mg, Mn, and Zn show analogous trends and high anomalies at sites S1 to S3 and S5, suggesting vehicle emissions and marble dust as major pollution sources [21, 128]. It has been reported that the main sources of K, Al, and Fe in surface water are potash fertilizers, weathering of potash silicate reserves, rainwater, and discharge of domestic sewage [38]. In addition, abundant concentrations of Na, P, and Sr in an aquatic environment reflect the integration of anthropogenic and crustal contributions [129].

## Conclusions

The current study assessed water quality for irrigation, heavy metal concentrations with associated human health risks, and identification of sources of pollution. The summarized results concluded that the surface water was heavily polluted by Al, Ca, Cr, Cu, Fe, K, Mg, Mn, Ni, P, and Zn and posed considerable carcinogenic and noncarcinogenic health hazards from ingestion and dermal routes to the target groups. Effluents from the marble industry are the main source of surface water pollution in the study area. Therefore, it is strongly recommended to move the marble processing plants from the nearby river canals (Swat River) and agricultural land to the industrial area. Regular monitoring to assess pollution levels in surface waters is needed, and the implementation of phytoremediation strategies is helpful to reduce pollution contributions to their default levels. For future improvements in the agricultural sector, the use of biochar and diatomite (chemical soil remediation agents) is recommended for the reclamation of heavy metal-contaminated soils. It is highly recommended to implement safe working practices such as sewage treatment plants and safety measures during irrigation and work in the marble industry without affecting the environmental resources. The findings of the current study are of great value to countries where marble runoff generally enters freshwater resources and is particularly helpful to the Government of Khyber Pakhtunkhwa, Pakistan, in formulating policies prohibiting industrial activities near and on freshwater bodies and agricultural land to protect aquatic and terrestrial life.

## Supporting information

**S1 Table. Detection limit (LOD), wavelength and % recovery of the analyzed elements.** (DOCX)

**S2 Table. Water quality indices with categories used in the study.** (DOCX)

**S3 Table. Parameters used for calculation of noncarcinogenic and carcinogenic dermal risk assessment.** (DOCX)

## Acknowledgments

The authors are thankful to Dr. Muhammad Daud Head of Central Analytical Facility Division at Pakistan Institute of Nuclear Science and Technology (PINSTECH) Islamabad for provision of laboratory facilities.

## Author Contributions

**Conceptualization:** Asghar Khan, Muhammad Saleem Khan.

**Data curation:** Asghar Khan, Juan José Egozcue, Ghulam Saddiq.

**Formal analysis:** Asghar Khan, Juan José Egozcue, Munib Ahmed Shafique, Sidra Nadeem.

**Investigation:** Asghar Khan.

**Methodology:** Asghar Khan, Muhammad Saleem Khan.

**Project administration:** Muhammad Saleem Khan.

**Resources:** Muhammad Saleem Khan, Munib Ahmed Shafique, Sidra Nadeem.

**Software:** Asghar Khan, Juan José Egozcue.

**Supervision:** Muhammad Saleem Khan.

**Validation:** Asghar Khan, Munib Ahmed Shafique, Sidra Nadeem, Ghulam Saddiq.

**Visualization:** Asghar Khan, Munib Ahmed Shafique, Sidra Nadeem, Ghulam Saddiq.

**Writing – original draft:** Asghar Khan.

**Writing – review & editing:** Muhammad Saleem Khan, Juan José Egozcue, Ghulam Saddiq.

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
