## [Decision Letter · Decision Letter 0]

20 Jun 2022

PONE-D-22-06684

Irrigation suitability and heavy metal screening of surface water near marble industry with allied human health risks in District Malakand, Pakistan

PLOS ONE

Dear Dr. khan,

Thank you for submitting your manuscript to PLOS ONE. After careful consideration, we have decided that your manuscript does not meet our criteria for publication and must therefore be rejected.

Based on the adverse comments received from the reviewers I decided to reject the submission at this stage.

I am sorry that we cannot be more positive on this occasion, but hope that you appreciate the reasons for this decision.

Kind regards,

Amitava Mukherjee, ME, Ph.D.

Academic Editor

PLOS ONE

Reviewers' comments:

Reviewer's Responses to Questions

**Comments to the Author**

1. Is the manuscript technically sound, and do the data support the conclusions?

Reviewer #1: No

2. Has the statistical analysis been performed appropriately and rigorously? 

Reviewer #1: No

3. Have the authors made all data underlying the findings in their manuscript fully available?

Reviewer #1: Yes

4. Is the manuscript presented in an intelligible fashion and written in standard English?

Reviewer #1: Yes

5. Review Comments to the Author

Reviewer #1: This manuscript surveyed the elemental composition in surface water for irrigation in the marble plant nearby. The study approach, findings, and discussion is local and not novel. Additionally, this reviewer can not agree with the concept of HI calculation and the involved element and the exposed way consideration. How is the suspended solid impact for human health by inhalation? The data findings were not new and local, it is hard to get new knowledge for readers.

6. PLOS authors have the option to publish the peer review history of their article (what does this mean?). If published, this will include your full peer review and any attached files.

Reviewer #1: **Yes: **Zeng-Yei Hseu

- - - - -

---

## [Author Response · Author response to Decision Letter 0]

9 Jul 2022

Reviewer comments

1. Is the manuscript technically sound, and do the data support the conclusions?

Answer: 

The manuscript is technically sound and the data support the conclusion very well. The study area is divided into three experimental units in the following way: (i) Control (C) (ii) Industrial sites (S1-S7) and downstream (S8). The data was collected in Replication with quite good sample size. The appropriated conclusion from the data sat was drawn and written in the manuscript

2. Has the statistical analysis been performed appropriately and rigorously?

 Answer: Yes. Descriptive and compositional data analysis tools were used for good interpretation of the data set.

3. Have the authors made all data underlying the findings in their manuscript fully available?

answer. Yes

4. Is the manuscript presented in an intelligible fashion and written in standard English?

answer. Yes

5. Review Comments to the Author.

Reviewer #1: This manuscript surveyed the elemental composition in surface water for irrigation in the marble plant nearby. The study approach, findings, and discussion is local and not novel. Additionally, this reviewer can not agree with the concept of HI calculation and the involved element and the exposed way consideration. How is the suspended solid impact for human health by inhalation? The data findings were not new and local, it is hard to get new knowledge for readers.

Answer.

• The reviewer agreed with presentation of the manuscript and the used language 

• The reviewer also agreed with Plos Data Policy and agree to acceptance of the manuscript

• Moreover, pollution of freshwater resource is a worldwide problem. Marble industry is operating worldwide generating waste in various forms discharge to agriculture fields or water bodies

• Previously little attention has been drawn to this industry as in most research articles use of marble contaminated water for irrigation is justified without the heavy metals composition and its toxic effects 

• In current study novel approaches were used to classify the usage of the water resource for irrigation of crops with associated health risks and novel statistical approach like compositional data analysis for better interpretation of the study results which were ignored in such types of study. 

• The HI is an accepted parameter used for evaluation of health risk related to many pathways. Moreover, the average daily adsorbed dose values were calculated, hazard quotient (HQ) values were calculated and finally the risk from heavy metal exposure through ingestion and dermal routes were calculated as HI which is a standard procedure followed in the High Impact Journals

• In our current study we evaluated the health risks through ingestion and dermal pathways

• There is no inhalation pathway as noted by the academic editor

• The suspended solid can have large impacts on human health through ingestion pathways by causing gastrointestinal issues but our main focus in the manuscript is retained to health risks due to heavy metals

• The data finding contains novelty approach like spatial mapping of heavy metals using ArC GIS, Compositional data approach, Kelly ratios, Principal component analysis approach and are helpful for those country which are facing scarcity of freshwater and their community use contaminated water for irrigation purposes without knowing the actual chemistry of the resource

---

## [Decision Letter · Decision Letter 1]

26 Aug 2022

PONE-D-22-06684R1Irrigation suitability, health risk assessment and source apportionment of heavy metals in surface water used for irrigation near marble industry in Malakand, PakistanPLOS ONE

Dear Dr. khan,

Thank you for submitting your manuscript to PLOS ONE. After careful consideration, we feel that it has merit but does not fully meet PLOS ONE’s publication criteria as it currently stands. Therefore, we invite you to submit a revised version of the manuscript that addresses the points raised during the review process.

ACADEMIC EDITOR: Please insert comments here and delete this placeholder text when finished. Be sure to:Indicate which changes you require for acceptance versus which changes you recommendAddress any conflicts between the reviews so that it's clear which advice the authors should followProvide specific feedback from your evaluation of the manuscriptPlease ensure that your decision is justified on PLOS ONE’s publication criteria and not, for example, on novelty or perceived impact.

We look forward to receiving your revised manuscript.

Kind regards,

Pramod K Pandey

Academic Editor

PLOS ONE

Journal Requirements:

Additional Editor Comments (if provided):

Dear Author:

Thank you for resubmitting this manuscript. After looking all the comments from multiple reviewers, and mixed responses, your manuscript was revaluated by new reviewers. Now we have outcomes, and a major revision has been requested. As an academic editor, I suggest authors to revisit manuscript, carefully look critical feedbacks, and provide line by line response to comments. Also, writing requires improvement in terms of language. Now this is an opportunity for authors to rework in manuscript, and resubmit a revision (improved version) of the manuscript for further review.

Thank you for considering PLOS ONE.

Regards

Academic Editor

Reviewers' comments:

Reviewer's Responses to Questions

**Comments to the Author**

1. If the authors have adequately addressed your comments raised in a previous round of review and you feel that this manuscript is now acceptable for publication, you may indicate that here to bypass the “Comments to the Author” section, enter your conflict of interest statement in the “Confidential to Editor” section, and submit your "Accept" recommendation.

Reviewer #2: (No Response)

Reviewer #3: (No Response)

2. Is the manuscript technically sound, and do the data support the conclusions?

Reviewer #2: Yes

Reviewer #3: Yes

3. Has the statistical analysis been performed appropriately and rigorously? 

Reviewer #2: Yes

Reviewer #3: Yes

4. Have the authors made all data underlying the findings in their manuscript fully available?

Reviewer #2: Yes

Reviewer #3: Yes

5. Is the manuscript presented in an intelligible fashion and written in standard English?

Reviewer #2: Yes

Reviewer #3: Yes

6. Review Comments to the Author

Reviewer #2: Alternative irrigation water sources are needed due to the decrease in available clean water resources. One of the best alternatives among these is the use of marginal waters such as wastewater. For this purpose, studies should be supported. In addition, field studies are important in terms of both time and effort. In the study, sampling, analysis and writing stages are sufficient.

Reviewer #3: Manuscript Number: PONE-D-22-06684R1

Manuscript Full Title: Irrigation suitability, health risk assessment and source apportionment of heavy metals in surface water used for irrigation near marble industry in Malakand, Pakistan

General comments

1. In this study, authors evaluated heavy metals (HM) levels in surface water nearby a marble industry for irrigation suitability purposes together with associated health risks in Malakand, Pakistan. Using GIS, authors collected 27 samples of surface water used for irrigation of crops and analyzed for heavy metals (Al, Ca, Cr, Cu, Fe, K, Mg, Mn, Na, Ni, P, Si, Sr, Ti, and Zn) through inductively coupled plasma optical emission. Using heavy metal contents, authors calculated health risks through ingestion and dermal pathway. Results of the Kelly’s ratio (KR) classified the surface water as unsuitable for irrigation. The average daily absorbed dose (ADD), total hazard quotient (THQ > 1) and hazard index (HI > 1) revealed considerable risk of noncarcinogenic toxicity for aduslts and children for certain heavy metals through the ingestion and dermal exposure routes. Especially, cancer risk factor values for Cr and Ni were also higher than the USEPA threshold revealing potential carcinogenic risk. Authors also performed multivariate analysis [CoDa PCA, PCA biplot (clr)]. Authors proposed regular monitoring and remediation strategies of studied waters before being used for irrigation.

2. Overall, manuscript contains useful data and can be a good contribution to scientific literature. Study plan is sound accompanied with the use of latest statistical analyses techniques such as PCA, which have made this study impressive. However, I recommend major revisions of manuscript based on following comments.

3. Manuscript needs language improvement. There are several sentences which contain either grammatical error or are not easy to follow/understand. For detail, please see specific comments.

Specific comments

4. Abstract needs revision. Currently, abstract is very general without any quantitative data. Methodological detail is missing in abstract. Authors may include number of samples collected in abstract. Authors may also present here the % of samples having heavy metal levels above limit values.

5. Introduction sections well-explains the study background and objectives. However, it needs language improvement. Moreover, authors may include some more data here about industrial water treatment and wastewater irrigation concept of the country. For reference, author may consult following review article about wastewater production, composition and use for irrigation in the country.

6. A critical analysis of wastewater use in agriculture and associated health risks in Pakistan. Environmental Geochemistry and Health. doi.org/10.1007/s10653-020-00702-3

7. A review of environmental contamination and health risk assessment of wastewater use for crop irrigation with a focus on low and high income countries. International Journal of Environmental Research and Public Health. 15(5), 895

8. Reference citation is vague and needs careful attention. Authors must follow journal style and format for citations. Currently, authors have cited article by numbers and name of authors. For example:

• Line 155: The people of the study area are primarily connected with agricultural sector that grows various cereal crops and vegetables (Wahid et al., 2017).

• Line 229: (Aitchison, 1982; Egozcue et al., 2003; Egozcue and Pawlowsky-Glahn, 2005).

• Line 153: According to (44)

9. Moreover, authors have cited a total of 142 references in this which are too many for a research article. Please cite only necessary reverences and limit the reference count to < 40.

10. Methodology is well-described. Study area is well-explained and all the risk indices are well-presented with proper references. However, Statistical analysis is too long and contains unnecessary detail and too many references.

11. What is the justification/reason of selecting the following heavy metals (metals (Al, Ca, Cr, Cu, Fe, K, Mg, Mn, Na, Ni, P, Si, Sr, Ti, and Zn) for analysis in this study? Why ot some other important heavy metal(loid)s such as As, Cd, Pb etc.

12. Results and discussion section seems ok. It would be better to concise this section if possible. The introductory sentences in each section, which are basically repetitions, may be deleted. For example, Line 461-462: “Assessment of human health risk is an approach widely used to compute the nature and probability of hostile health effects in humans (124).”.

13. Conclusion seems like abstract. Here, authors must present their concluding results and recommendations rather than just repeating abstract.

14. Authors have used several abbreviations in the manuscript. However, the use of abbreviations is not as per rule. Generally, an abbreviation must be used at the first appearance of full name in the text and then the abbreviation must be used throughout the manuscript. Here authors have used both randomly.

15. Following are some examples of sentences needing grammatical revision or are not easy to understand/follow. Entire manuscript needs grammatical revision.

• Line 22-23: Health risks from HM exposure were calculated through ingestion and dermal pathway by application …..

• Line 32-34: Estimated Incremental lifetime cancer risk values for Cr and Ni were also higher than the USEPA threshold of 1 x 10−6 - 1 x 10−4 revealed potential carcinogenic risk.

• Line : The current study results revealed both non-carcinogenic and carcinogenic health risks to the exposure population.

• Line 37-38: Therefore, regular monitoring and remediation strategies is ….

• Line 55: Industrialization and urbanization process is beneficial….

• Line 55-58: Too long sentence to follow. Please use small and easily understandable sentences: “Industrialization and urbanization process is beneficial, particularly in the aspect of shaping demographic characteristics, however the rapid, unplanned, random and uncontrolled conversion of the tangible landscape may result undesirable change on water and land ecosystem in terms of deterioration and scarcity of fresh water, posing an extensive threat to global food security (1–3).”

• Line 83-84: Globally the issue of heavy metal (HM) pollution of fresh water resources getting more attention due to the toxic, persistence and bioaccumulative nature (19)

• Line 153: According to (44) the supplementary contributor of pollutant load to the irrigation channels in study area were surface runoff, urban wastewater, and atmospheric deposition of heavy metals.

7. PLOS authors have the option to publish the peer review history of their article (what does this mean?). If published, this will include your full peer review and any attached files.

Reviewer #2: No

Reviewer #3: No

---

## [Author Response · Author response to Decision Letter 1]

15 Sep 2022

Response to reviewer comments

The reviewers’ comments on points 2 to 5 unanimously recommend the manuscript for acceptance. In addition, responses to the proposed changes by the reviewers from point six (6) onwards are discussed in detail in the proceeding paragraphs. 

2. Is the manuscript technically sound, and do the data support the conclusions?

Reviewer #2: Yes

Reviewer #3: Yes 

3. Has the statistical analysis been performed appropriately and rigorously?

Reviewer #2: Yes

Reviewer #3: Yes

4. Have the authors made all data underlying the findings in their manuscript fully available?

Reviewer #2: Yes

Reviewer #3: Yes

5. Is the manuscript presented in an intelligible fashion and written in standard English?

Reviewer #2: Yes

Reviewer #3: Yes

Reviewer #2:

6. Alternative irrigation water sources are needed due to the decrease in available clean water resources. One of the best alternatives among these is the use of marginal waters such as wastewater. For this purpose, studies should be supported. In addition, field studies are important in terms of both time and effort. In the study, sampling, analysis and writing stages are sufficient.

Answer: 

Using wastewater for irrigation is the best alternative source of irrigation. We discussed this alternative source of irrigation with a slight modification in the manuscript in lines (60-65:76-78:103-110:116-123). Reusing wastewater for irrigation alleviates the intense pressure on freshwater extraction and provides nutrients to plants and crops but without proper management (treatment), wastewater can affect plant growth, human health, general economic conditions of the community, and even the cultural and religious values of human society as wastewater contains pathogens (bacteria, viruses, protozoa, and molds) and potentially toxic elements in excessive concentrations. Moreover, the reviewer endorsed that the current study sampling, analysis, and writing stages are sufficient. 

Reviewer #3: 

General comments

1. In this study, authors evaluated heavy metals (HM) levels in surface water nearby a marble industry for irrigation suitability purposes together with associated health risks in Malakand, Pakistan. Using GIS, authors collected 27 samples of surface water used for irrigation of crops and analyzed for heavy metals (Al, Ca, Cr, Cu, Fe, K, Mg, Mn, Na, Ni, P, Si, Sr, Ti, and Zn) through inductively coupled plasma optical emission. Using heavy metal contents, authors calculated health risks through ingestion and dermal pathway. Results of the Kelly’s ratio (KR) classified the surface water as unsuitable for irrigation. The average daily absorbed dose (ADD), total hazard quotient (THQ > 1) and hazard index (HI > 1) revealed considerable risk of noncarcinogenic toxicity for aduslts and children for certain heavy metals through the ingestion and dermal exposure routes. Especially, cancer risk factor values for Cr and Ni were also higher than the USEPA threshold revealing potential carcinogenic risk. Authors also performed multivariate analysis [CoDa PCA, PCA biplot (clr)]. Authors proposed regular monitoring and remediation strategies of studied waters before being used for irrigation.

Answer: The reviewers' comments are clear and support the acceptance of the manuscript for publication.

2. Overall, manuscript contains useful data and can be a good contribution to scientific literature. Study plan is sound accompanied with the use of latest statistical analyses techniques such as PCA, which have made this study impressive. However, I recommend major revisions of manuscript based on following comments.

Answer: The reviewer's comments are clear and recommend the manuscript for publication with major revisions. The revision was made following the reviewers' comments and the responses were included in the next sections. 

3. Manuscript needs language improvement. There are several sentences which contain either grammatical error or are not easy to follow/understand. 

Answer: The manuscript has been completely revised; the grammatical errors have been rectified. The long sentences have also been revised.

4. Abstract needs revision. Currently, abstract is very general without any quantitative data. Methodological detail is missing in abstract. Authors may include number of samples collected in abstract. Authors may also present here the % of samples having heavy metal levels above limit values.

Answer: The abstract has been revised. Quantitative data in terms of samples and percent results are included. Methodological details were also revised and well explained. 

5. Introduction sections well-explains the study background and objectives. However, it needs language improvement. Moreover, authors may include some more data here about industrial water treatment and wastewater irrigation concept of the country. For reference, author may consult following review article about wastewater production, composition and use for irrigation in the country.

Answer: Language of the introduction section was improved. The data on industrial water treatment was included (lines: 111-115, 525-529) and wastewater irrigation concept of the country was also well explained (lines: 60-66, 103-109,116-123).

6. A critical analysis of wastewater use in agriculture and associated health risks in Pakistan. Environmental Geochemistry and Health. doi.org/10.1007/s10653-020-00702-3

7. A review of environmental contamination and health risk assessment of wastewater use for crop irrigation with a focus on low and high income countries. International Journal of Environmental Research and Public Health. 15(5), 895

Answer: About wastewater production, composition and use for irrigation in the country. The author fully read the above two articles and extracted the concept and presented well in the introduction section of the manuscript (lines 81-93;105-110)

8. Reference citation is vague and needs careful attention. Authors must follow journal style and format for citations. Currently, authors have cited article by numbers and name of authors. For example:

• Line 155: The people of the study area are primarily connected with agricultural sector that grows various cereal crops and vegetables (Wahid et al., 2017).

• Line 229: (Aitchison, 1982; Egozcue et al., 2003; Egozcue and Pawlowsky-Glahn, 2005).

• Line 153: According to (44)

Answer: The reference citations have been corrected according to the Journal style and format. The citation errors highlighted the lines number (line155; Line 229 and Line 153) have been reformatted. 

9. Moreover, authors have cited a total of 142 references in this which are too many for a research article. Please cite only necessary reverences and limit the reference count to < 40.

Answer: The author completely agrees with the reviewer's suggestions to reduction of references. The references were reduced to 129. All the cited references are very important and relevant that logically support the manuscript. Further reduction in references may lead to biased opinions. 

10. Methodology is well-described. Study area is well-explained, and all the risk indices are well-presented with proper references. However, Statistical analysis is too long and contains unnecessary detail and too many references.

Answer: The statistical analysis was revised and the unnecessary details with too many references were deleted.

11. What is the justification/reason of selecting the following heavy metals (metals (Al, Ca, Cr, Cu, Fe, K, Mg, Mn, Na, Ni, P, Si, Sr, Ti, and Zn) for analysis in this study? Why of some other important heavy metal(loid)s such as As, Cd, Pb etc.

Answer: The samples were analyzed by ICP-OES with a special focus on heavy metal(loid)s such as As, Cd, Pb as well. The metal(loid)s such as As, Cd, Pb was not detected by the machine and will include in the future study.

12. Results and discussion section seems ok. It would be better to concise this section if possible. The introductory sentences in each section, which are basically repetitions, may be deleted. For example, Line 461-462: “Assessment of human health risk is an approach widely used to compute the nature and probability of hostile health effects in humans (124).”.

Answer: The result and discussion section were concise and the highlighted repetitions in Lines 124 and 461-462 were deleted 

13. Conclusion seems like abstract. Here, authors must present their concluding results and recommendations rather than just repeating abstract.

Answer: The conclusion was revised, and the concluding results and recommendations were well presented (Line 515-533).

 14. Authors have used several abbreviations in the manuscript. However, the use of abbreviations is not as per rule. Generally, an abbreviation must be used at the first appearance of full name in the text and then the abbreviation must be used throughout the manuscript. Here authors have used both randomly.

Answer: Abbreviations in the manuscript have been rectified according to the Journal guidelines

15. Following are some examples of sentences needing grammatical revision or are not easy to understand/follow. Entire manuscript needs grammatical revision.

• Line 22-23: Health risks from HM exposure were calculated through ingestion and dermal pathway by application …..

• Line 32-34: Estimated Incremental lifetime cancer risk values for Cr and Ni were also higher than the USEPA threshold of 1 x 10−6 - 1 x 10−4 revealed potential carcinogenic risk.

• Line : The current study results revealed both non-carcinogenic and carcinogenic health risks to the exposure population.

• Line 37-38: Therefore, regular monitoring and remediation strategies is ….

• Line 55: Industrialization and urbanization process is beneficial….

• Line 55-58: Too long sentence to follow. Please use small and easily understandable sentences: “Industrialization and urbanization process is beneficial, particularly in the aspect of shaping demographic characteristics, however the rapid, unplanned, random and uncontrolled conversion of the tangible landscape may result undesirable change on water and land ecosystem in terms of deterioration and scarcity of fresh water, posing an extensive threat to global food security (1–3).”

• Line 83-84: Globally the issue of heavy metal (HM) pollution of freshwater resources getting more attention due to the toxic, persistence and Bio accumulative nature (19)

• Line 153: According to (44) the supplementary contributor of pollutant load to the irrigation channels in study area were surface runoff, urban wastewater, and atmospheric deposition of heavy metals.

Answer: The large sentences lines no. (22-23; 32-34; 37-38; 55; 55-58;83-84; and 153) have been completely revised, grammatically corrected and understandable.

---

## [Decision Letter · Decision Letter 2]

1 Dec 2022

Irrigation suitability, health risk assessment, and source apportionment of heavy metals in surface water used for irrigation near marble industry in Malakand, Pakistan

PONE-D-22-06684R2

Dear Dr. khan,

We’re pleased to inform you that your manuscript has been judged scientifically suitable for publication and will be formally accepted for publication once it meets all outstanding technical requirements.

Kind regards,

Pramod K Pandey

Academic Editor

PLOS ONE

Additional Editor Comments (optional):

Few corrections needed before publishing:

1. Authors should improve written English, or should take the help from, who has experience in scientific writing for improving English.

2. Abbreviation should be written in parenthesis instead they should follow the pattern which is common, and previously published in PlosOne.

3. Conclusion should be written as conclusions.

Above are few examples. Authors are suggested to spend substantial time on improving the English of manuscript, and follow the writing of PlosOne.

Reviewers' comments:

Reviewer's Responses to Questions

**Comments to the Author**

1. If the authors have adequately addressed your comments raised in a previous round of review and you feel that this manuscript is now acceptable for publication, you may indicate that here to bypass the “Comments to the Author” section, enter your conflict of interest statement in the “Confidential to Editor” section, and submit your "Accept" recommendation.

Reviewer #2: All comments have been addressed

Reviewer #4: All comments have been addressed

2. Is the manuscript technically sound, and do the data support the conclusions?

Reviewer #2: Yes

Reviewer #4: Yes

3. Has the statistical analysis been performed appropriately and rigorously? 

Reviewer #2: Yes

Reviewer #4: Yes

4. Have the authors made all data underlying the findings in their manuscript fully available?

Reviewer #2: Yes

Reviewer #4: Yes

5. Is the manuscript presented in an intelligible fashion and written in standard English?

Reviewer #2: Yes

Reviewer #4: Yes

6. Review Comments to the Author

Reviewer #2: The required corrections in the study were answered adequately. Publication of the study is appropriate.

Reviewer #4: Authors have appropriately addressed the reviewers comments. The layout of the manuscript is good and the presentation of the data is systematic. It can be consider for the publicaiton.

7. PLOS authors have the option to publish the peer review history of their article (what does this mean?). If published, this will include your full peer review and any attached files.

Reviewer #2: No

Reviewer #4: No

---

## [Editor Report · Acceptance letter]

12 Dec 2022

PONE-D-22-06684R2 

Irrigation suitability, health risk assessment and source apportionment of heavy metals in surface water used for irrigation near marble industry in Malakand, Pakistan 

Dear Dr. khan:

I'm pleased to inform you that your manuscript has been deemed suitable for publication in PLOS ONE. Congratulations! Your manuscript is now with our production department. 

Kind regards, 

on behalf of

Dr. Pramod K Pandey 

Academic Editor

PLOS ONE